# Integrated Intensified Chemoradiation in the Setting of Total Neoadjuvant Therapy (TNT) in Patients with Locally Advanced Rectal Cancer: A Retrospective Single-Arm Study on Feasibility and Efficacy

**DOI:** 10.3390/cancers15030921

**Published:** 2023-02-01

**Authors:** Maria Chiara Lo Greco, Madalina La Rocca, Giorgia Marano, Irene Finocchiaro, Rocco Luca Emanuele Liardo, Roberto Milazzotto, Grazia Acquaviva, Antonello Basile, Stefano Palmucci, Pietro Valerio Foti, Stefano Pergolizzi, Antonio Pontoriero, Silvana Parisi, Corrado Spatola

**Affiliations:** 1Department Scienze Biomediche, Odontoiatriche e delle Immagini Morfologiche e Funzionali, Università di Messina, 98122 Messina, Italy; 2U.O.S.D. Radioterapia Oncologica, A.O.U. Policlinico “G. Rodolico-San Marco” Catania, Via Santa Sofia 78, 95123 Catania, Italy; 3U.O.C. Radioterapia, A.O.E. Cannizzaro, 95126 Catania, Italy; 4U.O. Radiologia I, A.O.U. Policlinico “G. Rodolico-San Marco”, Via Santa Sofia 78, 95123 Catania, Italy; 5Department Scienze Mediche, Chirurgiche e Tecnologie Avanzate “G.F. Ingrassia”, Università di Catania, 95123 Catania, Italy; 6U.O.C. Radioterapia Oncologica, A.O.U. Policlinico “G. Martino” Messina, Via Consolare Valeria 1, 98125 Messina, Italy

**Keywords:** locally advanced rectal cancer, total neoadjuvant therapy, long course radiotherapy, chemoradiotherapy, conventionally fractionated radiotherapy, hypofractionated radiotherapy, intensified preoperative chemoradiotherapy

## Abstract

**Simple Summary:**

Based on literature data suggesting promising advantages of total neoadjuvant therapy (TNT) in patients with locally advanced rectal cancer, we performed a retrospective single-arm, single-center study on 45 patients affected by histologically and radiologically proven locally advanced rectal cancer, with the aim of analyzing the feasibility and short-term efficacy of an integrated intensified treatment, including induction chemotherapy, concurrent chemoradiation with long-course radiotherapy, and concomitant boost and consolidation chemotherapy. At a median follow-up of 30 months, this strategy has shown to be feasible and effective in terms of pathological complete response (pCR) and short-term disease-free survival (DFS).

**Abstract:**

While surgery is considered the main treatment for early-stage rectal cancer, locally advanced rectal cancer needs to be handled with a multidisciplinary approach. Based on literature data suggesting promising advantages of total neoadjuvant therapy (TNT), we performed a retrospective, single-arm, single-center study on 45 patients affected by histologically and radiologically proven locally advanced rectal cancer, with the aim of analyzing the feasibility and short-term efficacy of an integrated intensified treatment in the setting of TNT. Each analyzed patient performed three cycles of FOLFOX4 or De Gramont induction chemotherapy (iCT), followed by concurrent chemoradiotherapy (CRT) with long course radiotherapy (LCRT) plus concomitant boost and continuous 5-FU infusion, followed by three cycles of FOLFOX4 or De Gramont consolidation chemotherapy (conCT) and then surgery with total mesorectal excision. At a median follow-up of 30 months, this strategy has shown to be feasible and effective in terms of pathological complete response (pCR) and short-term disease-free survival (DFS).

## 1. Introduction

Colorectal cancer is the third most common cancer in western countries and the second leading cause of cancer death in the world [1].

During the last few decades, the outcome for patients with rectal cancer has significantly improved in many high-income countries, due to the development of various treatment strategies.

In the early 1980s, a better understanding of rectal surgical oncology led to the introduction of a new operative technique, called total mesorectal excision (TME), a specific surgical approach consisting in the complete removal of the entire circumferential perirectal tissue envelope. Due to the clinical practice of TME, the 5-year survival improved from 45–50% to 75%, local recurrence rates decreased from 30% to 5–8%, the sphincter preservation for mid- and lower rectal cancer increased by 20%, and impotence and bladder dysfunction rates declined from 50–85% to 15% [2,3].

In the following years, multiple trials investigating the role of neoadjuvant therapies in the management of locally advanced rectal cancer (LARC) assessed the advantages of delivering preoperative chemoradiotherapy (CRT) to improve local control rates and increase sphincter preservation rates in patients with low-lying tumors [4,5,6].

Despite the current standard of care is combined-modality therapy withneoadjuvant CRT, surgery with TME and adjuvant chemotherapy, distant metastasis remains the major reason for treatment failure. In the aim to reduce the risk of distant recurrence, several studies investigated the potential advantage of a novel approach, termed total neoadjuvant therapy (TNT), which consists of delivering systemic chemotherapy in the neoadjuvant setting, either before or after CRT. Early results have demonstrated high rates of completion and tolerability, pathological complete response (pCR), sphincter-saving surgery, and R0 resection. Moreover, the current pooled data has shown TNT benefit in the rate of disease-free survival (DFS) and 3-year overall survival (OS) [7].

Based on literature data suggesting promising advantages of TNT strategy, we performed a retrospective, single-arm, single-center study with the aim of analyzing the feasibility and toxicity rates of an integrated intensified treatment in the setting of TNT. Moreover, we investigated short-term TNT efficacy, in terms of pCR rates and DFS.

## 2. Materials and Methods

At the University of Catania, we retrospectively analyzed 45 patients affected by histologically and radiologically proven locally advanced rectal cancer, from 2017 to 2021.

After a careful history and physical examination, including digital rectal exam, all patients underwent an endoscopic examination with rigid sigmoidoscopy both to measure the distance from the lesion to the anal verge and to pathologically confirm rectal cancer diagnosis.

All patients underwent pelvic magnetic resonance imaging (MRI) to determine local tumor extension and nodal status and computed tomography (CT) of the brain, chest, abdomen, and pelvis to identify metastatic lesions. Then, they were all staged according to UICC TNM staging (8th edition) classification [8].

We selected 45 patients according to the following inclusion criteria: tumor location within 15 cm of the anal verge, histology of adenocarcinoma, locally advanced clinical stage with absence of metastasis, written informed consent, and performance status > 80% (Karnofsky scale).

All patients were candidates for an integrated intensified treatment in the setting of TNT, which consisted of three cycles of FOLFOX4 or De Gramont induction chemotherapy (iCT), followed by concurrent CRT with long course radiotherapy (LCRT) plus concomitant boost and continuous 5-fluorouracil (5-FU) infusion, followed by three cycles of FOLFOX4 or De Gramont consolidation chemotherapy (conCT) and then surgery with TME.

After surgery, all patients underwent trimestral follow-up for the first year, and then semestral follow-up from the second year onward. At each visit, all patients were clinical and hematological evaluated. Moreover, they underwent semestral instrumental evaluation with CT scan (brain, chest, abdomen, and pelvis) or colonscopy.

To determinate the feasibility and toxicity rates, all patients were clinical and hematological evaluated both during iCT, CRT and conCT. Toxicity data were collected according to Common Terminology Criteria for Adverse Events (CTCAE v4.0) [9].

To determine short-term efficacy, we analyzed pCR rates after surgery, approximately 8 weeks after the end of TNT. Then we assessed DFS, at a median follow-up of 30 months.

### 2.1. Patients’ Data

From 2017 to 2021, a total of 45 patients were selected to be analyzed in this study; 30 patients were males and 15 were females. The patients’ ages ranged between 43 and 85 years (median = 70 years). Clinical presentation was mostly characterized by different onset symptoms, such as rectal bleeding and abdominal pain (44.4%), change in bowel habits or irregular alvus (40%), tenesmus (37.7%), and sub-obstruction symptoms (26.6%).

After endoscopic examination with rigid sigmoidoscopy, 18 patients were found to be affected by low rectal cancer (<6 cm), 22 patients by middle rectal cancer (6.1–10 cm), and 5 patients by high rectal cancer (>10 cm).

At pretreatment TNM staging (cTNM), 7 patients were cT2 cN+, 8 patients were cT3-T4 cN0, 30 patients were cT3-4 cN+. The characteristics of the patients are listed in Table 1.

### 2.2. Induction, Concurrent and Consolidation Chemotherapy Schemes

Prior to the start of iCT, all patients received a peripherally inserted central catheter (PICC), and then, to avoid severe adverse events related to fluoropyrimidine treatment, the activity of the DPD enzyme was tested [10]. All patients were found to have wild-type DPYD. Before every chemotherapy infusion, premedication with dexamethasone (8 mg) and ondansetron (8 mg) was administered.

ICT was performed according to the following schemes:-FOLFOX4 regimen, based on Oxaliplatin 85 mg/m^2^ intravenous (IV) infusion (day 1) and leucovorin 200 mg/m^2^ IV infusion administered concurrently over 120 min, followed by 5-FU 400 mg/m^2^ IV bolus, followed by 5-FU 600 mg/m^2^ IV infusion over 22 h (day 1 and 2) every two weeks. The FOLFOX4 induction regimen was administered to 35 patients.-De Gramont regimen, based on leucovorin 200 mg/m^2^ IV infusion over 120 min (day 1 and 2) followed by 5-FU 400 mg/m^2^ IV bolus (day 1 and 2), followed by 5-FU 600 mg/m^2^ IV infusion over 22 h (day 1 and 2), leucovorin 400 mg/m^2^, every two weeks. The De Gramont induction regimen was administered to 10 patients.

The choice of which chemotherapy schedule to perform was tailored based on patients’ characteristics. The De Gramont regimen was preferred in very elderly patients or patients with two or more comorbidities, to avoid severe toxicity related to oxaliplatin.

Afterwards, each one of the selected patients received CRT with LCRT concurrent with fluoropyrimidine based chemotherapy.

Concurrent chemotherapy was performed according to the following schemes:-5-FU continuous infusion regimen, based on 5 FU 225 mg/m^2^ IV infusion over 24 h daily (1–5 or 1–7), every week for five weeks. The 5-FU continuous infusion regimen was administered to all patients.

About two weeks after the last radiotherapy fraction, conCT was performed following the same schemes as iCT.

-The FOLFOX4 consolidation regimen was administered to 32 patients.-The De Gramont consolidation regimen was administered to 13 patients.

### 2.3. Radiotherapy Treatment Planning and Delivery

To design a treatment plan unique for each patient’s anatomy, after an adequate bowel preparation, every patient underwent a radiotherapy simulation process through an abdomen-pelvis CT scan without contrast enhancement.

Since the patient’s position must be the same for both initial simulation and subsequent treatment, during this phase an optimal set-up was needed: all patients were placed in the prone position using a belly board device, used to displace the small bowels out of the treatment fields.

CT images were acquired with 3–5 mm slice spacing, from L1 to femur half, then data were analyzed using treatment planning software (XIO) for target volume definition and dose solutions, according to our institutional protocol [11].

During the contour delineation phase, a gross tumor volume (GTV) was outlined, including the primary tumor and enlarged regional nodes, and the clinical target volume (CTV) was outlined including the GTV, rectum, mesorectum and posterior pelvic subregions. The mesorectum, presacral, obturator, and internal iliac lymph nodes were included in all patients’ treatment volumes, while the external iliac lymph nodes were included only if clinically positive or in case of T4 tumor.

A planning target volume (PTV) was obtained, giving an expansion of approximately 5 mm (three dimensional) to CTV, to account for daily set-up error and organ motion.

The organ at risk (OARs) were bowel (Dax < 55 Gy), bladder (V50 60%, V60 50%), femoral heads (V50 60%) and anal canal (Dmax < 55 Gy).

Radiation therapy was delivered by means of a Siemens ONCOR linear accelerator using a 3D-conformal technique (3D-CRT) at 6–15 MV or a static step-and-shot intensity-modulated technique (IMRT). IMRT was preferred to avoid intestinal adhesions in patients with previous abdominal surgery and to avoid a high dose to the small bowel with a 3DCRT treatment plan (V15 > 120 cc or V45 > 195 cc of small bowel) [12].

During the entire course of radiotherapy treatment, patients were set up daily by using one sagittal and two lateral tattoos and lasers. A megavoltage cone-beam computer tomography image-guided radiotherapy (MV CBCT IGRT) system was used two times a week to check patients’ setups.

Different fractionation schedules were used, including conventionally fractionated radiation therapy (CFRT) or hypofractionated radiation therapy (HFRT). Total dose and dose per fraction were established considering an α/β ratio of 5.06 Gy, as suggested by literature data [13].

-CFRT consisted of a total dose of 45–46 Gy in 23–25 fractions (1.8–2 Gy/day) delivered to PTV. During the last week of treatment, a boost of 8–9 Gy in 4–5 fractions (1.8–2 Gy/day) was delivered to GTV, so that this volume received a total dose of 54 Gy.

CRFT was administered to 33 patients.

-HFRT consisted of a total dose of 40–42.75 Gy in 16–19 fractions (2.25–2.5 Gy/day) delivered to PTV. During the last week of treatment, a boost of 9–10 Gy in 4 fractions (2.25–2.5 Gy/day) was delivered to GTV. So that this volume received a total dose of 50–51.75 Gy.HFRT was administered to 12 patients.

The characteristics of treatment are listed in Figure 1.

### 2.4. Patients’ Evaluation

To determine the feasibility and toxicity rates of the TNT approach, before every iCT and conCT cycle and once a week during the entire course of CRT, all patients were clinical and hematological evaluated to identify adverse events. Toxicities were defined following the Common Terminology Criteria for Adverse Events (CTCAE) Version 4.0.

All patients underwent pelvic MRI and CT scans of the brain, chest, abdomen, and pelvis at the time of diagnosis to be staged, and again about 6–7 weeks after the end of TNT, to analyze radiological response. No re-evaluation imaging was planned at the end of iCT phase.

Radiological response was evaluated according to Response Evaluation Criteria in Solid Tumors (RECIST 1.1) [14].

To improve the diagnostic performance of MRI to evaluate the tumor response to TNT, high-resolution T2-weighted and diffusion-weighted imaging (DWI), with qualitative and quantitative evaluation through the measurement of tumors; apparent diffusion coefficient (ADC) were used, according to our institutional protocol [15,16].

After surgery, which was performed approximately 8 weeks after the end of TNT, pathological complete response rates were analyzed.

The imaging and the post-operative staging were then compared to evaluate the correspondence between the clinical and pathological assessments.

To assess the benefits in terms of disease-free survival, after surgery, all patients underwent trimestral follow-up for the first year, and semestral follow-up from the second year after surgery onward. Median follow-up was 30 months.

## 3. Results

### 3.1. Feasibility and Toxicity of Integrated Intensified TNT

During iCT and conCT, the majority of patients presented mild adverse events related to oxaliplatin or fluoropyrimidines (Figure 1). Due to poor clinical tolerance, we decided to switch three patients receiving FOLFOX4 iCT to De Gramont conCT.

Among patients performing iCT or conCT with FOLFOX4 regimen, five patients (11.1%) manifested G1 anemia and decreased platelet count; three patients (6.6%) experienced G1 hypertransaminasemia, lymphopenia and neutropenia; four patients (8.8%) experienced constipation and fatigue; and three patients (6.6%) experienced fatigue, nausea, and vomiting. ICT dose modification was performed in four patients (8.8%) due to G2 G.I. toxicity (nausea and vomiting, diarrhoea). ICT interruption (1–2 weeks) was performed in two patients (4.4%) due to G2 anemia and decreased platelet count. Of all patients receiving iCT and conCT with the FOLFOX4 regimen, five patients (11.1%) experienced G1 peripheral neuropathy.

Among patients receiving iCT or conCT with the De Gramont regimen, three patients (6.6%) manifested G1 anemia and decreased platelet count. ICT dose modification was performed in two patients (4.4%) due to G2 decreased platelet count.

All the chemoradiotherapy schemes were well-tolerated in the majority of patients.

During the course of CRT, five patients (11.1%) and three patients (6.6%) manifested G1 and G2 radiation dermatitis, respectively; three patients (6.6%) undergoing CFRT, and 1 patient (2.2%) undergoing HFRT, had their treatment interrupted (from 2 to 7 days) due to G3 diarrhea and proctitis. Moreover, four patients (8.8%) manifested G1 anemia and decreased platelet count.

The majority of patients (66.6%) experienced symptomatic relief during or after TNT. No significant differences between the different chemotherapy regimens or radiotherapy schedules were detected.

Regarding chronic toxicities, after 18 months, only three patients (6.6%) experienced peripheral neuropathy, and four patients (8.8%) experienced G1 proctitis.

### 3.2. Efficacy of Integrated Intensified TNT

About 6–7 weeks after the end of TNT, all patients underwent pelvic MRI and CT scans of the brain, chest, abdomen, and pelvis. Clinical TNM after neoadjuvant therapies (ycTNM) was evaluated according to Response Evaluation Criteria in Solid Tumors (RECIST 1.1) and then compared to pathological TNM (ypTNM) after surgery (as shown in Figure 2):-Partial response (ycPR) was reported in 24 patients (53.3%).-Complete response (ycCR)was reported in 8 patients (17.7%).-Stable disease (ycSD) was reported in 13 patients (28.8%).

Surgery was performed approximately eight weeks after the end of TNT. A low anterior resection with sphincter preservation was performed in 37 patients, and a temporary diverting loop ileostomy was performed in 25 patients. Miles abdomino-perineal resection was performed in eight patients. The results of all patients (100%) were R0.

The response obtained after TNT allowed surgeons to perform sphincter-saving surgery in 10 out of 18 patients initially selected for abdomino-perineal resection (distal tumour location < 6 cm).

The only post-surgical complications observed were perianastomotic fistula and intestinal obstruction (two patients).

Pathological TNM after surgery (ypTNM) was compared to ypTNM, and the response was:-ypPR in 27 patients (60%).-ypCR in 10 patients (22.2%).-ypSD in 8 patients (17.7%).

The pCR rates were consistent with literature data supporting the TNT strategy (Table 2) [17].

After surgery, every patient continued trimestral follow-up for the first two years, and semestral follow-up after that. At each visit, patients were clinical and hematological evaluated. Moreover, they performed semestral instrumental evaluation with CT scan (brain, chest, abdomen and pelvis) or colonscopy. To date, after a median follow-up of 30 months, 43 out of 45 patients (95.5%) are still alive, and two patients (4.4%) died of cardiovascular comorbidities without evidence of tumor relapse. Regarding DFS rates, seven patients (15.5%) experienced progression disease (PD); three patients (6.6%) were diagnosed with hepatic metastasis after a median follow-up of 5.5 months; three patients (6.6%) were diagnosed with lung metastasis after a median follow-up of 12 months; and one patient (2.2%) had local relapse (presacral) after 13 months (Figure 3).

## 4. Discussion

While surgery is considered the main treatment for early-stage rectal cancer, locally advanced rectal cancer needs to be handled with a multidisciplinary approach.

Since the late 1990s, the role of neoadjuvant radiotherapy has been investigated. First, the efficacy of preoperative short-course radiotherapy (SCRT) to reduce rates of local recurrence and improve survival was demonstrated by the Swedish Rectal Cancer Trial and the Dutch Trial. Then, The German CAO/ARO/AIO-94 Trial established preoperative chemoradiotherapy with 5-FU, TME surgery, and postoperative chemotherapy with 5-FU as standard treatment for LARC [4,5,6].

Currently, different evidence-based approaches are supported for LARC treatment. Concurrent CRT and LCRT is generally performed in the United States and several European countries. It delivers 45–50 Gy of external-beam, intensity-modulated radiation therapy over 25 to 28 daily fractions, with concomitant radiation-sensitizing, fluoropyrimidine-based chemotherapy such as oral capecitabine, continuous infusion of 5-FU or bolus 5-FU with leucovorin, followed by delayed surgery 6–8 weeks later. SCRT is preferred in Northern European countries. It aims to sterilize the mesorectal fat before surgery, and the regimen does not include chemotherapy. Patients receive 5 Gy per day for a total of 5 days, then typically proceed to surgery within 1–2 weeks [18,19,20,21].

Surgery is performed according to different techniques, such as sphincter-saving low anterior resection (LAR), or non-sphincter-saving abdomino-perineal resection (APR), based on preoperative staging. Both approaches include TME to ensure margins and lymph node retrieval [22].

In the postoperative setting, additional chemotherapy can be performed according to different schemes, such as: FOLFOX4 (oxaliplatin, 5-FU, and leucovorin) regimen, 5-FU and leucovorin, CAPOX (capecitabine plus oxaliplatin), or capecitabine alone [23]. Even if the use of adjuvant chemotherapy is recommended by different international guidelines, its use in patients who undergo preoperative CRT is controversial. Results of recent multicentre randomised control trials showed no benefit in terms of survival and distant metastases rates. Therefore, its use routinely requires rigorous evaluation [24,25,26].

The administration of this multimodal approach accounted for reduction in the local pelvic recurrence rates to less than 10%, but the possibility of distant metastasis remains more than twice that of primary tumour recurrence; furthermore, it is not free from complications. Literature data suggest that because of poor postoperative physical condition, more than a third of the patients delay or refuse adjuvant chemotherapy, and less than half of the eligible patients receive the full course of chemotherapy [27].

In the interest of improving compliance rates, reducing toxicity, and decreasing distant relapse rates, multiple prospective trials investigated the advantages of shifting systemic therapy delivery earlier in the treatment paradigm, incorporating induction or consolidation chemotherapy and chemoradiotherapy in the neoadjuvant setting, supporting the administration of a novel approach termed total neoadjuvant therapy.

The theoretical considerations which provide a solid rationale for TNT are: the potential to optimally treat preoperative occult micrometastatic disease, the possibility of delivering chemotherapy agents directly to the primary tumour while it has a fully intact vasculature (undisrupted by radiation or surgery), the prospect of reducing the duration of temporary ostomies, and avoiding the challenges of chemotherapy with an ostomy, improving the quality of life after surgery [28,29].

On these bases, the value of this approach has been recently investigated [30,31,32]. In 2020, TNT has been recommended as acceptable for stage II and III LARC by the National Comprehensive Cancer Network (NCCN) [33].

In 2021, the first two large, randomised phase III trials comparing TNT with the standard approach, in both cases CRT, released their preliminary results. The RAPIDO trial compared SCRT, followed by CAPOX or FOLFOX4 consolidation chemotherapy, followed by surgery (without adjuvant treatment) with CRT, followed by surgery and CAPOX or FOLFOX4 adjuvant chemotherapy. The PRODIGE-23 trial compared modified FOLFIRINOX followed by CRT, surgery and capecitabine or modified FOLFOX6 adjuvant chemotherapy with CRT, followed by surgery and capecitabine or modified FOLFOX6 adjuvant chemotherapy [34,35,36].

The primary endpoint was 3-year disease-related treatment failure for the RAPIDO trial and 3-year disease-free survival for PRODIGE 23. Both studies met their respective primary endpoints with statistically significant hazard ratios, paving the way for a new practical approach [37,38,39].

Currently, total neoadjuvant therapy can be performed following three different strategies: chemoradiotherapy followed by consolidation chemotherapy; induction chemotherapy followed by chemoradiotherapy; and induction chemotherapy, followed by chemoradiotherapy, followed by consolidation chemotherapy (sandwich TNT) [40,41].

To date, the most-performed induction and consolidation chemotherapy schemes are capecitabine-based, or 5-FU based. The role of bevacizumab, a monoclonal antibody against vascular endothelial growth factor (VEGF), has been lately investigated; however, due to the small amount of relevant data, further studies are needed [42,43,44].

A systematic review published in 2021 reported that patients receiving induction chemotherapy type TNT are 28% more likely to have complete pathological response, 44% less likely to have residual nodal disease at surgery, and 57% less likely to have positive margins, while patients receiving consolidation chemotherapy type TNT are 90% more likely to have complete pathological response, even if it does not provide a better nodal downstaging, and has similar positive margin rates to standard long course CRT. Sandwich TNT has been demonstrated to be highly effective in terms of pCR and major regression [45,46,47].

Basing on literature data suggesting promising advantages of TNT strategy, we performed a retrospective, single-arm, single-center study on 45 patients affected by histologically and radiologically proven locally advanced rectal cancer, with the aim of analyzing the feasibility and toxicity rates of an integrated intensified treatment in the setting of TNT. Moreover, we investigated short-term TNT efficacy, in terms of pCR rates and DFS.

All patients were candidates to receive three cycles of FOLFOX4 (35 patients) or De Gramont (10 patients) induction chemotherapy, followed by concurrent CRT with long-course radiotherapy plus concomitant boost and continuous infusion of 5-FU (45 patients), followed by three cycles of FOLFOX4 (32 patients) or De Gramont (13 patients) consolidation chemotherapy.

Surgery was performed approximately eight weeks after the end of TNT.

The response obtained after TNT allowed surgeons to perform sphincter-saving surgery in 10 out of 18 patients initially selected for abdomino-perineal resection (distal tumour location < 6 cm).

All the chemoradiotherapy schemes were well tolerated; the majority of patients presented mild adverse events related to oxaliplatin or fluoropyrimidines.

The PCR rates (22.2%) and DFS at a median follow-up of 30 months (80%) are consistent with literature data and support the efficacy of TNT in terms of short-term outcomes. Limitations of this study are that the selected population is relatively poor, and a longer median follow-up is needed to evaluate the real advantages of this strategy.

## 5. Conclusions

Our treatment protocol, based on integrated intensified treatment in the setting of total neoadjuvant therapy, has shown to be feasible and well tolerated, with only mild adverse events related to oxaliplatin or fluoropyrimidines.

Even if it is currently only possible to evaluate short-term outcomes, the high rate of pCR (22.2%) and DFS at a median follow-up of 30 months (80%) is encouraging. However, long-term follow-up is still required to determine if these advantages translate into improved overall survival.

## Figures and Tables

**Figure 1 cancers-15-00921-f001:**
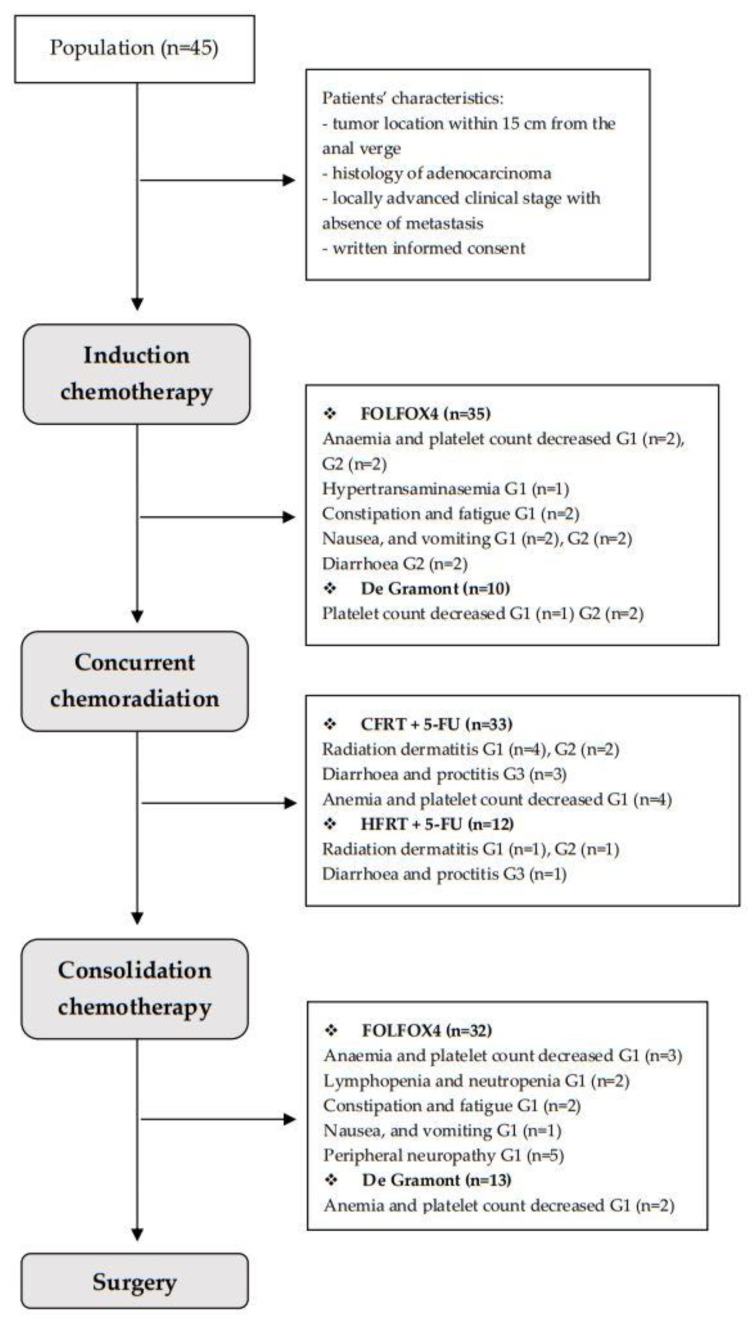
Consort diagram showing the different application and feasibility of preoperative induction chemotherapy (FOLFOX4, De Gramont) radio-chemotherapy (CFRT, HFRT), and consolidation chemotherapy (FOLFOX4, De Gramont).

**Figure 2 cancers-15-00921-f002:**
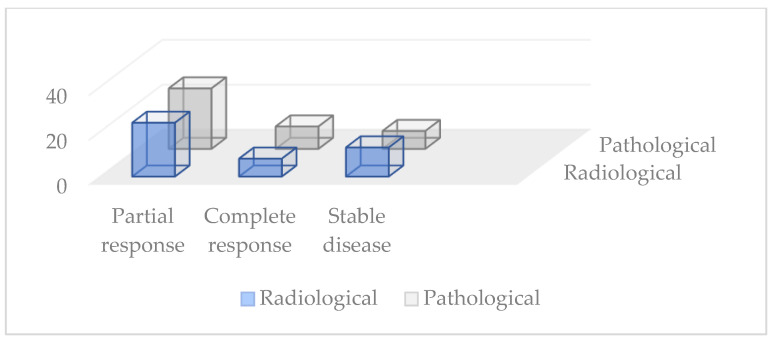
Comparation between pathological and radiological response.

**Figure 3 cancers-15-00921-f003:**
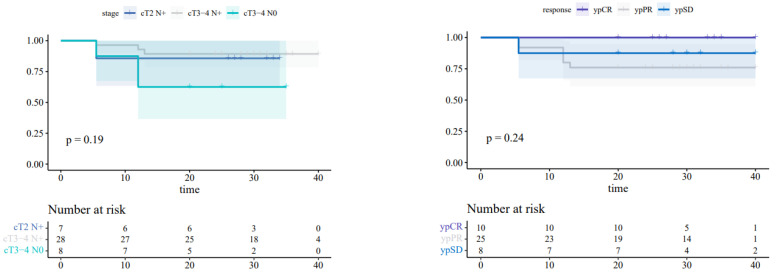
Kaplan–Meier curves showing DFS rates comparing different risk groups (cTNM, patological response after neoadjuvant therapy).

**Table 1 cancers-15-00921-t001:** Patients’ characteristics.

Characteristics		N. of Patients	Percentage
Sex	Males	30	66.6%
	Females	15	33.3%
Age	<54 y.o.	7	15.5%
	55–64 y.o.	12	26.6%
	65–74 y.o.	10	22.2%
	>75 y.o.	16	35.5%
KPS	100%	27	60%
	90%	14	31.1%
	80%	4	8.8%
Comorbidities	None	11	24.4%
	Diabetes	11	24.4%
	Hypertension	26	57.7%
	Hypercholesterolemia	6	13.3%
Onset symptoms	Rectorrhagia	20	44.4%
	Abdominal pain	20	44.4%
	Irregular alvus	18	40%
	Tenesmus	17	37.7%
	Substruction/obstruction	12	26.6%
Grading	G1	10	22.2%
	G2	15	33.3%
	G3	20	44.4%
Distance from mesorectal fascia	>1 mm	31	68.8%
	<1 mm	14	31.1%
Distance from A.O.	<6 cm	18	40%
	6.1–10 cm	22	48.8%
	>10 cm	5	11.1%
Stage	T2 N1-2	7	15.5%
	T3-T4 N0	8	17.7%
	T3-T4 N1-2	30	66.6%

**Table 2 cancers-15-00921-t002:** Literature data supporting the TNT strategy [17].

Source	No. of Events/Total	Percentage
Calvo et al. 2006	15/52	28.8%
Cerek et al. 2018	110/308	35.7%
Conroy et al. 2020	64/231	27.7%
Garcia-Aguilar et al. 2015	25/65	38.4%
Markovina et al. 2017	19/69	27.5%
Van der Valk et al. 2020	117/423	27.6%
Van Zoggel et al. 2018	10/58	17.24%
Our Study	10/45	22.2%

## Data Availability

The data presented in this study are available on request from the corresponding author. The data are not publicly available due to privacy restrictions.

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
