# Peer review of "Integrated Intensified Chemoradiation in the Setting of Total Neoadjuvant Therapy (TNT) in Patients with Locally Advanced Rectal Cancer: A Retrospective Single-Arm Study on Feasibility and Efficacy"

_cancers, 2023, doi:10.3390/cancers15030921_

Round 1

Reviewer 1 Report

not all patients were locally advanced, also cT2 tumours were included

how were patients selected? patient preference? doctors preference?

is there a study protocol approved by the medical ethical committee?

during follow-up: was this combined with imaging (CT, MRI?) at every visit? or only clinicallly with/without CEA? imaging of course can influence DFS in a important way

what's the interval between last RT fraction and start of conCT?

was a watch&wait stategy allowed in case of ycCR?

result result section is not very extensive. more information on surgical outcome would be welcome. R0/R1 resection (margins) percentage is missing.

no information about long term tox of chemo,

regarding DFS: can this be provided in kaplan meier curves?

how was recurrence detected? at which location? how about local recurrence?  especially interested since more LARs were performed .

Author Response

As shown in table 1, all cT2 tumours are N+ (N1 or N2).

In our centre, TNT is proposed to all patients affected by LARC with PS >80% (KS). All patients are informed about risks and benefits.

This is a retrospective study, so a protocol number was not necessary.

Patients have been clinically and hematologically evaluated at each visit, moreover they performed semestral CT scan or colonscopy. I added this specification.

All patients started conCT about two weeks after the last RT fraction. I added this specification.

We retrospectively selected patients who performed surgery. In our centre we hold watch&wait strategy only to patients who can’t be subjected to surgery for important comorbidities.

All patients were R0, I have added this specification in 3.2 section.

I added the specification about chronic toxicities.

I provided two Kaplan Meier curves comparing different risk groups.

Each patients performed semestral instrumental evaluation with CT scan or endoscopy. Only one patient had presacral recurrence. I added this specification.

Reviewer 2 Report

This is a retrospective study of 45 patients treated at a single center with undefined inclusion criteria, treated heterogeneously (chemotherapy including oxaliplatin in some patients and not in others). The population included in the analysis is also heterogeneous (cT2-cT4). The choice of TNT treatment (sandwich strategy) is not justified in the manuscript. Furthermore, the study has important methodological flaws such as the lack of a statistical design nor does it adequately define survival outcomes.  For all these reasons, I consider that this study does not add anything relevant to the current knowledge of the treatment of locally advanced rectal cancer.

Author Response

As the other reviewers suggested, I added some specification including a consort diagram and two Kaplan Meier curves in the hope of giving more significance to the study.

The treatment schedules are heterogenous because we generally tailor TNT strategy basing on patients' characteristics (I added this specification in the text).

Unfortunately, the selected population is small, but we hope to obtain more relevant results in the future increasing the number of patients and prolonging the follow-up.

Reviewer 3 Report

The authors analyse the results of a retrospective analysis of a single arm, single center study on 45 patients with locally advanced rectal cancer who received preoperative induction chemotherapy with intensified chemoradiotherapy followed by consolidative chemotherapy (TNT).

The results are interesting with regard to the endpoints of feasibility, toxicity, pathological response, and disease-free survival at a median follow-up of 30 months.

For a better understanding of the risk criteria of disease and the study results due to TNT, the following critical comments should be considered:

ad 1. Despite being a single arm study, a consort diagram should illustrate the different application of preoperative induction chemotherapy (FOLFOX4 ,De Gramont) radio-chemotherapy (CFRT,HFRT), and consolidation chemotherapy( FOLFOX4,De Gramont) with the number of patients who received or did not receive these therapeutic interventions. This replaces Tab. 2

ad 2. The characteristics of the patients (Table 1) are insufficient for comparison with the results of other studies. In addition to age, gender and performance, the characteristics of the patients should include Table1:

Clinical T and N category cT2,cT3,cT4; cN0, cN1-2

Clinical disease stage Stage II,Stage III

cT1-2 N1-2;cT3-4 N1-2 

Distance of tumor to mesorectal fascia, mm

<1 and >1

Location from anal verge

0-5,>5-10,>10

Histology: Adenocarcinoma ,others

Tumor differentiation: G1,G2,G3

ad 3.The feasibility is shown by the consort diagram (see comment ad1.). A key result is the accurate representation of toxicity. There is a significant deficit in the manuscript.

The authors should provide a common overview of the degree of toxicity of the therapy (CTCAE) divided into degrees 1-2, >degree 3 and the individual description of degree 5 (fatal) in a new table. The haematological toxicity, gastrointestinal side effects, neurological side effects (for example, hand-foot syndrome, oxaliplatin related neurotoxicity) and others should be included.(number of patients;% of patients).

ad.4 The results for cCR,cPR and pCR,pPR should be given with the results of surgical resection.

ad.5 Figure 2 /DFS does not correspond to the usual presentation of a Kaplan Meier survival curve. In addition to the number of patients at risk over the period of observation, this curve should also show a comparison of risk groups.

 General comments:

Limitations of the study should be given in the text of discussion.

The reasons of the relatively low pCR rate compared with other studies (see table 3) should be discussed.

Author Response

I provided a consort diagram showing the different applications and feseabilities of iCT, CRT and conCT, in the hope that is intuitive and easy to understand.

I added the characteristics you suggested in table 1, except for histology, since they were all adenocarcinoma.

Moreover, I provided two Kaplan Meier curves comparing different risk groups (cTNM at diagnosis and response after neoadjuvant treatment).

Thank you very much for your suggestions. 

Round 2

Reviewer 1 Report

reference 18 and 19 are inappropriate self-citations.

authors don't state whether they have  informed consent . 

it's hard to believe that all resections are radical (R0) since this study involves LARC . 

the follow-up schedule can be more specified. how often was CT scan performed? every year? it's good that a coloscopy is performed but this wont't detect metastases. maybe it can detect a Local recurrence or second primary in a low number of cases.

Reviewer 2 Report

This retrospective case report lacks interest due to the lack of a minimally acceptable methodology.

Reviewer 3 Report

The revised version now fulfills the proposed requirements and answers the open questions.